# Preparation and Basic Properties of Praseodymium-Neodymium-Chromium Containing Imitation Gemstone Glass

**DOI:** 10.3390/ma15207341

**Published:** 2022-10-20

**Authors:** Siqi Zhang, Keqing Li, Junyuan Pu, Wen Ni

**Affiliations:** 1School of Civil and Resource Engineering, University of Science and Technology Beijing, Beijing 100083, China; 2National Gems & Jewelry Testing Group Training Center Shanghai Branch, Shanghai 200021, China

**Keywords:** gem-imitation, rare earth glass, colour change effect, spectral characteristics, praseodymium-neodymium-chromium

## Abstract

Imitation gemstone glass has numerous characteristics, including low cost, rich colour, stable colouring, and the formation of colour-changing effects that can meet the jewellery market demand for beautiful gemstones of middle and low grades. In this study, four types of gem-imitating glass were prepared by the elemental substitution of praseodymium, neodymium and chromium elements based on rare earth glass and examined by combining refractive index, density, spectral characteristics and colour parameters. Sample 1 contained only Pr_6_O_11_ and showed a golden-yellow colour like chrysoberyl. Sample 2 contained only Nd_2_O_3_ and showed a blue-purple colour like amethyst. Sample 3 contained Pr_6_O_11_ and Nd_2_O_3_ and appeared green under D65 light source and red under A light source, with a colour-change effect like alexandrite. Sample 4 contained Pr_6_O_11_, Nd_2_O_3_ and Cr_2_O_3_ and showed a highly saturated green colour like emerald because of the strong colouring effect of Cr^3+^ in the glass. The findings revealed that all four samples are transparent, with a refractive index greater than 1.5 and a density higher than 2.6 g/cm^3^. The comprehensive performance of the four imitation gemstone glasses can be found in the corresponding natural gemstones, which has a certain practical value.

## 1. Introduction

Natural gemstones are exceedingly valuable precious mineral resources that are scarce and non-renewable. Imitation gemstone glass is an outstanding material for imitating gemstones, which has numerous advantages, including low cost and performance comparable to natural gemstones [1]. Previous research on imitation gemstone glass has focused on glass transparency and the emulsion effect [2]. In the past, studies on imitation gemstone glass focused on glass transparency and opalescence, such as several studies on lead-based glass [3,4]. Due to its environmental issues, researchers have been looking for alternative materials. Currently, the research mainly focuses on the diversification of glass colour as the primary goal while considering the characteristics of a high refractive index and strong luster.

The medieval artisan Heraclius was working on making beautiful shiny stones from Roman glass to imitate precious stones, such as hyacinths, green sapphires and gems of other colours [5]. Furthermore, several of these gems on reliquaries, book-covers and other objects in museums and cathedral treasure have been proven to be glass, including the Sapphire of Queen Theodolinda [6]. Rare earth elements are highly efficient in glass colouration, producing various colour shades [7]. They cover the colours of common gemstones. According to Weilin Huang’s study [8], most rare earth elements appropriate for glass colourants generally have poor colouring strength, but praseodymium’s colouring performance is outstanding. Furthermore, it is difficult to add praseodymium oxide to glass as a pure oxide alone, so it must be mixed with neodymium oxide for further colouring and to enhance the light transmission of incident light. According to Dingkun Shen’s study [9], a single chromium element exists in glass as Cr^3+^ (3d^3^), which is in the octahedral coordination structure to split the energy level and undergo spin-allowed leap, producing a strong and wide light transmission band. Lanthanum coronal glass is a high refractive index, high dispersion glass system. Its refractive index value is usually 1.65–1.75, the dispersion value is very high, and the dispersion coefficient (Abbe number) can reach 50–60 [10]. According to Shuyan Gao’s research [11], lanthanide rare earth elements such as cerium (Ce), praseodymium (Pr), neodymium (Nd), samarium (Sm), and dysprosium (Dy) can be employed as stable colourants to prepare imitation gemstone glass, and the combination of rare earth elements can also produce a colour change effect. Jialin Zhao [12] developed imitation gemstone glasses that show light red-purple, pink-yellow, and blue-violet colour change effects under fluorescent and incandescent lamps. Xiaoyan Yu [13] identified a type of rare earth imitation gemstone glass containing a trace amount of praseodymium and neodymium elements. It has a colour-changing effect similar to that of natural Turkish zultanite. However, there is still little research on imitation gemstone glass, and more research is required in the face of the enormous demand for gem-imitation products in the future market.

This study aims to develop glass varieties that could be used to imitate coloured gemstones. First, it was necessary to identify suitable glass materials for imitation gemstones that meet the performance requirements of a high refractive index, high dispersion, high density, and solid transparency, while ensuring a green and environmentally friendly production process. Additionally, the colouring effect of these colorants in imitation gemstone glass was clarified through ratio tests with different additions of praseodymium oxide, neodymium oxide and chromium oxide, combined with analytical means such as spectral data and colour parameters. Furthermore, the raw material source, preparation process, and the cutting method of processing the imitation gemstone glass into faceted gemstones were clarified.

## 2. Raw Materials and Methods

### 2.1. Experimental Raw Materials and Proportions

The purpose of imitation gemstone glass is to serve as a substitute for natural gemstones. The typical characteristics of natural gemstones are: (1) transparency; (2) rich colour; (3) hardness, with a Mohs hardness of 5 or more; (4) strong lustre, with a refractive index value of 1.50 or more; (5) density that can reach 2.6 g/cm3; (6) a certain dispersion intensity, which can produce a fire colour similar to diamonds. Based on these, the performance of imitation gemstone glass focuses on optical properties, especially the high refractive index and high dispersion, as the main goal of glass composition system selection. According to the book “Glass Technology”, lanthanum corona glass belongs to La_2_O_3_-SiO_2_-B_2_O_3_-BaO as the main system of lanthanum-barium-borosilicate glass, belongs to the high refractive index and has a high dispersion glass system. The refractive index value is usually 1.65~1.75, the dispersion value is very high, and the dispersion coefficient (Abbe number) can reach 50~60.

In this study, the lanthanum corona rare earth glass system was based on the addition of alkali metal oxides to enhance the flux effect. Four glass systems with SiO_2_-K_2_O-Na_2_O-B_2_O_3_-Al_2_O_3_-ZnO as the base glass, praseodymium oxide (Pr_6_O_11_), neodymium oxide (Nd_2_O_3_) and chromium oxide (Cr_2_O_3_) as the colouring agent were identified after extensive pre-screening experiments, and four of the most representative formulations were selected, as shown in Table 1. The primary raw materials were quartz sand (SiO_2_), alumina (Al_2_O_3_), sodium nitrate (NaNO_3_), potassium carbonate (K_2_CO_3_), borax (Na_2_B_4_O_7_·10H_2_O) and zinc oxide (ZnO). Sample 1 was coloured with a single praseodymium element. Sample 2 was coloured with a single neodymium element. Sample 3 was co-coloured with praseodymium. Sample 4 was co-coloured with praseodymium, neodymium, and chromium.

#### 2.1.1. Preparation Process

The molar composition of the glasses is shown in Table 1. The glass components included silicate dioxide (SiO_2_), aluminium oxide (Al_2_O_3_), sodium nitrate (NaNO_3_), potassium carbonate (K_2_CO_3_), borax (Na_2_B_4_O_7_·10H_2_O), and zinc oxide (ZnO), which were weighed out to give approximately 500 g of glass for each composition. The oxide powders were well mixed in a plastic container and transferred to a corundum crucible. The crucible was then placed in an electric furnace for a period of 1.5 h at a temperature of 1450 °C. Then, the glass liquid was poured into the iron shaping abrasives, quickly placed into the insulation electric furnace for annealing at an annealing temperature of 500 °C, insulated for 2 h and then allowed to cool in the furnace chamber where the glass specimen was made (Figure 1).

The cutting process of the glass determines the colour effect, brightness and fire of the imitation gemstone glass. Glass specimens must be cut according to the size specification requirements to become imitation gemstone glass with various facets that maximises the optical effect. The specific process of faceted imitation gemstone glass cutting included cutting → shaping → gluing → grinding → polishing. The prepared imitation gemstone samples were employed for subsequent testing (Figure 2).

#### 2.1.2. Characterisation of Imitation Gemstone Glass Properties

The basic characteristics of natural gemstones are transparency; colour; strong lustre and a refractive index value above 1.50; a density of 2.6 g/cm^3^ or more. Thus, the properties of transparency, colour, density, refractive index and fluorescence effect are crucial indicators to reflect whether the test sample can achieve a gemstone grade effect. In this research, a gem refractometer, FABLE FGR-002A, with a test approach of faceting, a range of 1.35–1.81, and a refractive index value retained to three decimal places, was employed. Other instruments included a hydrostatic weighing device, Shanghai Puchun JY2002; an ultraviolet fluorescent lamp, and FABLE FUV-4, that radiatde long-wave ultraviolet light with a main wavelength of 365 nm and a short-wave UV light with a wavelength of 253.7 nm. The chemical composition of the glass was accurately tested using iCAP TQ LA-ICP-MS (Inductively Coupled Plasma Mass Spectrometer) from Thermo Fisher, Waltham, MA, USA. The spectral transmission curves of the imitation gemstone glass in the wavelength range of 370–750 nm were measured using an X-Rite Ci7000A benchtop high-precision spectrophotometer for analysis and comparison. The visible spectral absorption characteristics of various samples were examined and characterised by the CIE chromaticity diagram and CIE Lab colour space, the former can express the actual colour of the glass accurately and quantitatively, and the latter can express the colour difference of the glass precisely.

## 3. Results

### 3.1. Study of Basic Physical Properties

The difference in chemical composition gives the imitation gemstone glass different property characteristics in terms of basic physical properties such as density, refractive index, and colour. Table 2 shows the imitation gemstone glass samples’ colour, refractive index, density, and fluorescence effect.

Samples 1 to 4 were cut and processed according to the degree of light transmission, transparency, and a refractive index greater than 1.5, and gradually increased in density, all greater than 2.6 g/cm^3^. As illustrated in Figure 3, samples 1, 2 and 4 in artificial daylight and incandescent light show yellow, blue-violet and emerald-green, respectively, with outstanding brightness and fire, sample 3 in artificial daylight shows yellow-green, in incandescent light showed brownish red, indicating a green hue to a large change in a red hue. Sample 1 had weak purplish-red fluorescence, sample 2 had no fluorescence, and samples 3 and 4 had moderate purplish-red fluorescence. Four samples met the performance requirements of imitation gemstones.

### 3.2. Optical Performance

A Perkin Elmer Lambda 950 UV-Vis spectrophotometer was used to test four samples using a transmission method in the wavelength range of 320–800 nm with a slit width of 1 nm. The transmission spectra of each sample were plotted, as shown in Figure 4. The chromaticity coordinates can be measured by spectrophotometer to determine the spectral curve and then calculated according to Equations (1) and (2). Equations (1) and (2) for the calculation of chromaticity coordinates are shown below.
Figure 4Transmission spectra of samples: (**a**) sample 1; (**b**) sample 2; (**c**) sample 3; and (**d**) sample 4.
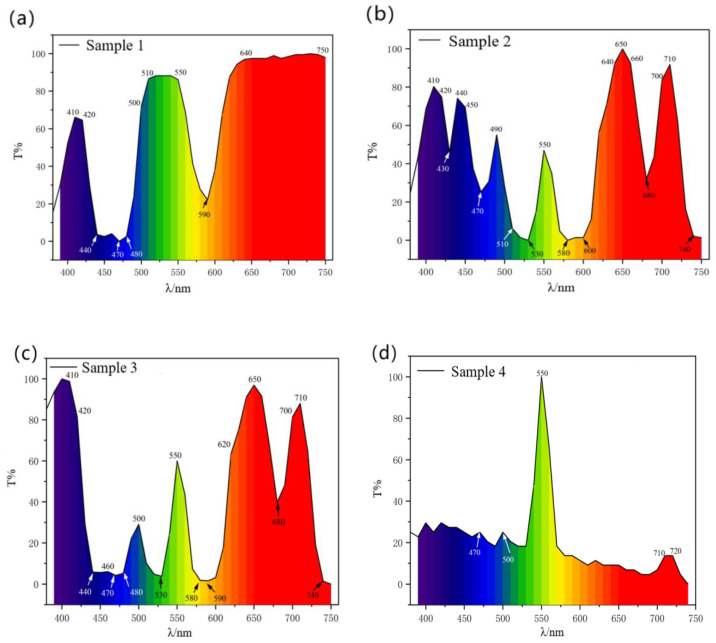

(1)$$\begin{array}{l} x=\frac{X}{X+Y+Z}\\ y=\frac{Y}{X+Y+Z}\\ z=\frac{X}{X+Y+Z} \end{array}$$

x, y, z—Chromaticity coordinates.*X*, *Y*, *Z*—The stimulus values of the three primary colours.



(2)
$$\begin{array}{l} X=\sum (H_{\lambda }\bar{x_{\lambda }}\tau_{\lambda }\Delta \lambda )\\ Y=\sum (H_{\lambda }\bar{y_{\lambda }}\tau_{\lambda }\Delta \lambda )\\ Z=\sum (H_{\lambda }\bar{z_{\lambda }}\tau_{\lambda }\Delta \lambda ) \end{array}$$



$H_{\lambda }\bar{y_{\lambda }}$—Wavelength of lighting source$\tau_{\lambda }$—The spectral transmittance of the glassΔλ—5 nm or 10 nm

The transmittance of the glass at different wavelengths is measured using a spectrophotometer and its chromaticity coordinates are calculated.

As illustrated in Figure 4, the transmission spectra of four samples were obtained by an ultraviolet visible spectrophotometer test. From the transmission spectrum in Figure 4a, in sample 1, praseodymium in the glass has typical transmission characteristics: there is a narrow transmission band in the blue purple region (410–420 nm), with an average transmission intensity of 65%. There is a strong transmission peak in the green region (510–550 nm), with a transmission intensity of about 90%. The whole red region (640–750 nm) is almost completely transmitted. According to the transmission spectrum in Figure 4b, in sample 2, neodymium in the glass has a plurality of transmission peaks or transmission bands. There are three transmission peaks in the ultraviolet region (410 nm, 440 nm, and 490 nm), with transmission intensities of 80%, 75% and 55%, respectively. There is a transmission band peak in the green region (550 nm), and the transmission intensity is 45%. There are two strong transmission peaks in the red region (640–660 nm and 700–710 nm), with transmission intensities of 100% and 90%. According to the transmission spectrum in Figure 4c, in sample 3, the transmitted light waves of praseodymium and neodymium appear at the same time, which enhance or cancel each other to produce a new colouring effect. The spectrum shows the superposition of Pr^3+^ transmission curve and Nd^3+^ transmission line, which strengthens the transmitted light waves in the purple region (410–420 nm) and the green region (550 nm), and weakens the transmitted light waves in the blue region (440–490 nm) and the red region (680 nm). Therefore, sample 3 is green in natural light and reddish brown in incandescent light, which has a colour-change effect similar to that of natural Alexandrite. According to the transmission spectrum in Figure 4d, in sample 4, the transmission characteristics of praseodymium, neodymium and chromium mainly show the typical spectral characteristics of chromium: the green region (550 nm) had a strong transmission peak, and the transmission intensity reached 100%. The light wave transmission intensity in other regions was relatively average, and the large transmission intensity was 10–20%.

As shown in Figure 5, Table 3 and Table 4, the spectral absorption data of the imitation gemstone glass samples were computed, and the colour parameters under the CIE chromaticity system and the CIE Lab colour space system were obtained.

Figure 5 illustrates the CIE chromaticity diagrams obtained from the spectral data of samples 1 to 4 after the formula calculation, which visually reveal the colour types of each sample.

In Table 2, the chromaticity coordinates provide a clear quantitative representation of the samples’ actual colour. The colour coordinates of sample 1 under the CIE 1931 colour characterisation system show that the actual colour of praseodymium tinted glass is a golden yellow consisting of 40.1% red, 47.6% green and 12.3% blue, which is very close to natural chrysoberyl’s colour. Under the CIE 1931 colour characterisation system, the colour coordinates of sample 2 show that the colour of the neodymium-coloured glass is a blue-violet consisting of 31.4% red, 20% green and 48.6% blue, which is very similar to the colour of natural amethyst. Both praseodymium and neodymium alone are very efficient in colouring glass, with Pr^3+^ effectively absorbing blue-violet and orange light to produce a golden-yellow colour similar to that of chrysoberyl, and Nd^3+^ effectively absorbing green and yellow light to produce a blue-violet colour similar to that of amethyst. The glass sample 3 employed to imitate the alexandrite is less red than green under D65 light source (6500 K) and appears yellow-green. The red proportion is significantly larger than the green proportion under the A light source (2856 K) and appears red. The actual colour of sample 4 is a vivid green composed of 28% red, 39.9% green and 32.1% blue, which is very similar to the colour of the natural emerald.

The CIE Lab colour space (Figure 6) is another colour system for characterising colours. The spectral curve is measured using a spectrophotometer, and the luminosity (*L**), chromaticity (a*, b*), saturation (*C**) and hue angle (h) for each sample were obtained by computation.

The spatial equation of CIE Lab is shown in Equation (3) below. *L** indicates lightness, a* and b* indicate chromaticity, *C** indicates saturation, h_0_ indicates hue angle, *X*, *Y* and *Z* indicate colour tri-stimulus values, and *X*_0_, *Y*_0_ and *Z*_0_ indicate colour tri-stimulus values of reference white.
(3)$$\begin{array}{l} L*=116\times {\left(\frac{Y}{Y_{0}}\right)}^{\frac{1}{3}}-16\\ L*=903.3\times \frac{Y}{Y_{0}}\\ a*=500\left[{\left(\frac{X}{X_{0}}\right)}^{\frac{1}{3}}-{\left(\frac{Y}{Y_{0}}\right)}^{\frac{1}{3}}\right]\\ b*=200\left[{\left(\frac{X}{X_{0}}\right)}^{\frac{1}{3}}-{\left(\frac{Z}{Z_{0}}\right)}^{\frac{1}{3}}\right]\\ C*={\left[{\left(a*\right)}^{2}+{\left(b*\right)}^{2}\right]}^{\frac{1}{2}}\\ h_{0}=\tan^{-1}(\frac{b*}{a*})=\arctan(\frac{b*}{a*}) \end{array}$$

The change in hue angle (h_0_) in the CIE Lab colour space is often employed to evaluate colour changes quantitatively. A significant colour change can be observed with the naked eye when the difference in the hue angle is greater than 20° [14]. Sample 3, employed to imitate the natural alexandrite, had a significant colour change effect under different light sources, Δh_0_ up to 45.42°, significantly greater than 20°, and the rest of the samples were less than 20°.

## 4. Discussion

In this research, SiO_2_-B_2_O_3_-Na_2_O-K_2_O-Al_2_O_3_-ZnO was employed as the primary component, and praseodymium oxide, neodymium oxide, and chromium oxide were employed as colourant additions to prepare four rare earth glass systems with high refractive index >1.5, high density >2.6 g/cm^3^ and high dispersion characteristics.

The production of glass colour results from the joint action of glass and light. The praseodymium and neodymium chromium elements exist in the glass structure in the form of Pr^3+^, Nd^3+^ and Cr^3+^. “Spectroscopy of Rare Earth Ions” [15] and “Mineralogical Applications of Crystal Field Theory” [16] contain detailed accounts of the spectral properties and spectral theories of various ions. Since the outermost 4f electron shell layer of praseodymium and neodymium ions and the outermost 3d electron shell of chromium ions are typically unfilled, under the irradiation of an external light source the 4f orbital electrons undergo an f-f jump between energy levels and the 3d orbital electrons undergo a d-d jump between energy levels, absorbing some specific wavelengths of light and producing specific. Simultaneously, the unabsorbed transmitted light enters the human eye, stimulating the human eye to produce colour, forming the colouring and colour-changing effect of praseodymium-chromium glass.

The transmission spectrum of chrysoberyl is characterised by a broad transmission band at 430 nm and a broad transmission band at 550 to 630 nm [17]. As shown in Figure 7, the spectral transmission characteristics of natural chrysoberyl are the same as those of sample 1. From the standpoint of the transmission spectrum, sample 1 can imitate the chrysoberyl very well. A broad transmission band characterises the transmission spectra of amethyst at 345 nm and a broad transmission band at 545 nm [18]. The spectral transmission features of sample 2 are mostly the same as those of natural amethyst, and from the standpoint of the transmission spectra, sample 2 can imitate natural amethyst very well. A strong transmission band characterises the transmission spectrum of natural alexandrite at wavelengths of 405–440 nm, a broad transmission band at wavelengths of 565–590 nm, and a transmission peak at wavelengths of 680 nm [19]. The spectral transmission characteristics of sample 3 are the same as those of the natural alexandrite, which can be a good imitation of the natural alexandrite from the viewpoint of transmission spectra. A broad transmission band characterises the transmission spectrum of natural emerald in the purple region at wavelengths of 420–430 nm and a broad transmission band in the orange-red region at wavelengths of 600–700 nm [20]. The spectral transmission features of sample 4 and the natural emerald are the same, and from the standpoint of transmission spectra, sample 4 can imitate the natural emerald well.

The colouring effect of Cr^3+^ is so prominent that the original colouring by three elements, praseodymium and chromium, becomes a separate colouring effect by chromium ions. According to the study on the mechanism of ion colouring in glass in “Introduction to Color Glass” [21], chromium ions are subjected to a change in electronic energy level under the action of oxygen ion electric field which eventually affects the spectral properties. Cr^3+^ is surrounded by O^2^^−^, and an octahedral coordination field is formed, according to the coordination field theory. As illustrated in Figure 8, the d orbital, an unsaturated electron orbital, is split into low-energy orbital groups (dxy, dyz and dxz) and two groups of high-energy orbital groups (d_z_^2^ and d_x^2^ − y^2^_) by the action of the oxygen ion electric field.

As shown in the Figure 8, d_e_ is the ground state of Cr^3+^, d_r_ is the high-energy excited state, and electrons leap between d_e_~d_r_. The d-d electron leap just makes the blue-violet light (380–470 nm) and yellow light (560–590 nm) absorbed, which finally leaves the green light (500–560 nm) to pass through the sample. Thus, the praseodymium-chromium glass appears to be a bright green colour similar to the natural emerald in the transparent state.

However, natural gemstones are generally non-fluorescent or weakly fluorescent under long-wave or short-wave UV fluorescence irradiation. However, fluorescence tests on prepared glass samples indicate that both praseodymium and neodymium single element colouring, praseodymium-neodymium combined colouring and praseodymium-chromium three element colouring reveal weak-moderate purplish-red fluorescence under 365 nm long-wave UV light. This makes the imitation effect defective, and the fluorescence effect becomes a crucial basis for identifying natural gemstones or imitation gemstone glass. Thus, the formulation design of future imitation gemstone glass must consider the introduction of fluorescence inhibitors to achieve a better imitation effect.

## 5. Conclusions

In this research, the glass ratios of single praseodymium glass, single neodymium glass, praseodymium glass and praseodymium-chromium glass were designed using praseodymium oxide (Pr_6_O_11_), neodymium oxide (Nd_2_O_3_), and chromium oxide (Cr_2_O_3_) as colourants.

All four imitation gemstone glasses are transparent, with a refractive index higher than 1.5, and a density up to 2.6 g/cm^3^. Under D65 and A light sources, sample 1 is yellow, sample 2 is blue-purple, sample 3 is yellow-green and brownish red with a colour-change effect, and sample 4 is vivid green.

The transmission band in the blue-violet region (440–480 nm) and the transmission peak in the orange region (590 nm) is the characteristic transmission spectrum of Pr^3+^. The characteristic transmission of praseodymium-chromium glass is the superposition of the transmission curves of Pr^3+^ and Nd^3+^, and the characteristic transmission of praseodymium-chromium glass is the superposition of the transmission curves of Pr^3+^, Nd^3+^. The characteristic transmission of praseodymium-chromium glass is the superposition of the transmission curves of Cr^3+^ due to the colouring effect of Cr^3+^.

## Figures and Tables

**Figure 1 materials-15-07341-f001:**
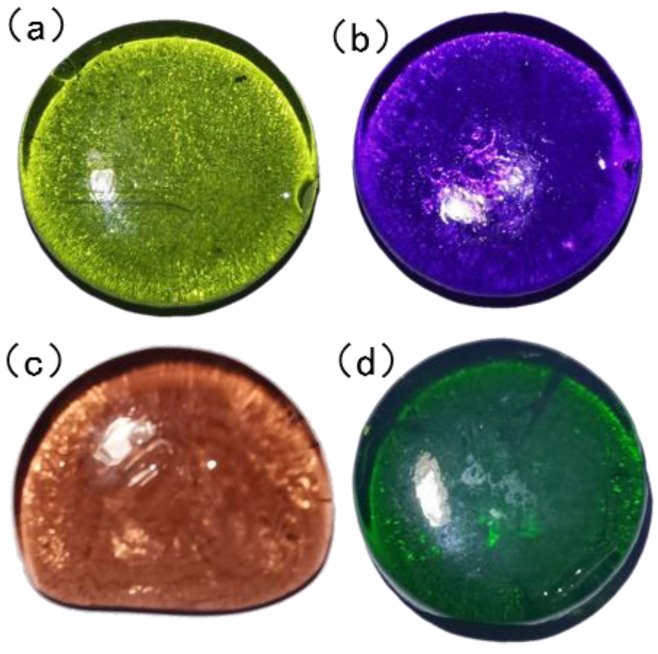
Imitation gemstone glass: (**a**) sample 1; (**b**) sample 2; (**c**) sample 3; (**d**) and sample 4.

**Figure 2 materials-15-07341-f002:**
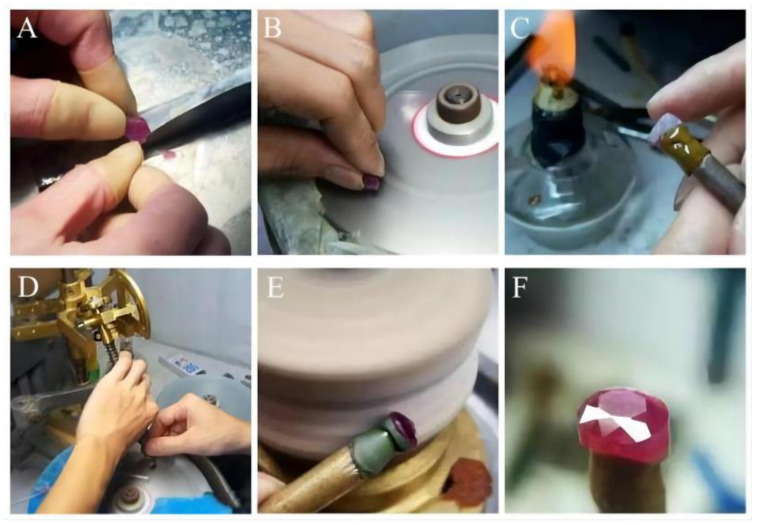
Process of hand-cut faceted imitation gemstone glass: (**A**) opening; (**B**) shaping; (**C**) gluing; (**D**) grinding; (**E**) polishing; and (**F**) final product.

**Figure 3 materials-15-07341-f003:**
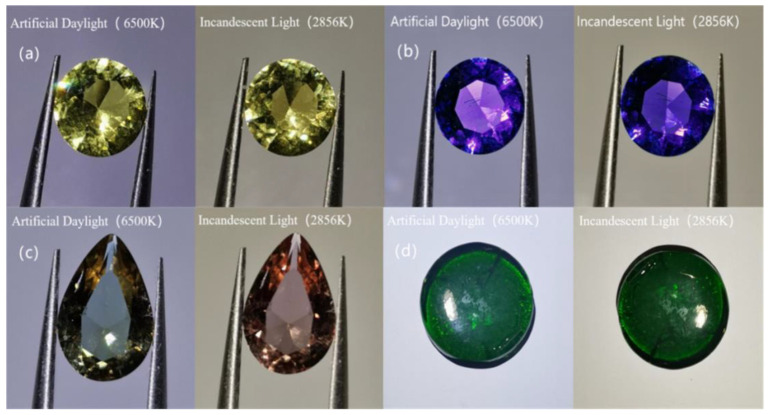
Samples in artificial daylight and incandescent lamp: (**a**) sample 1; (**b**) sample 2; (**c**) sample 3; and (**d**) sample 4.

**Figure 5 materials-15-07341-f005:**
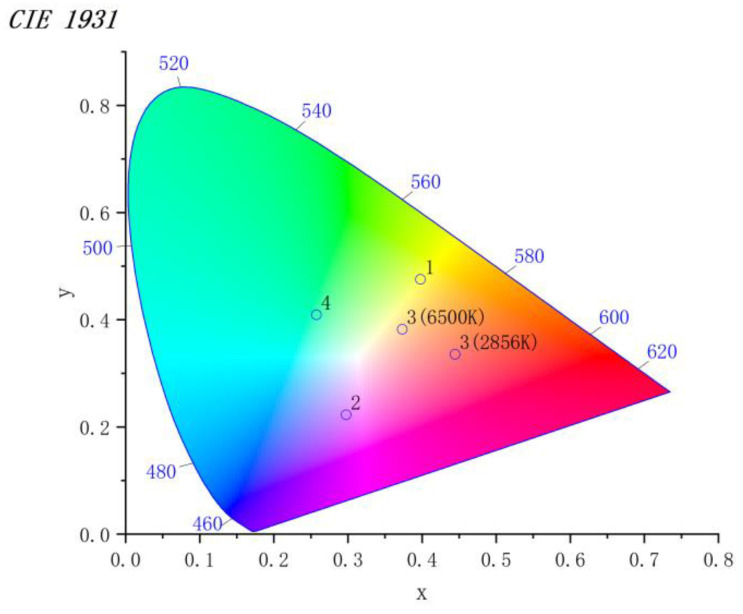
CIE chromaticity diagram of samples.

**Figure 6 materials-15-07341-f006:**
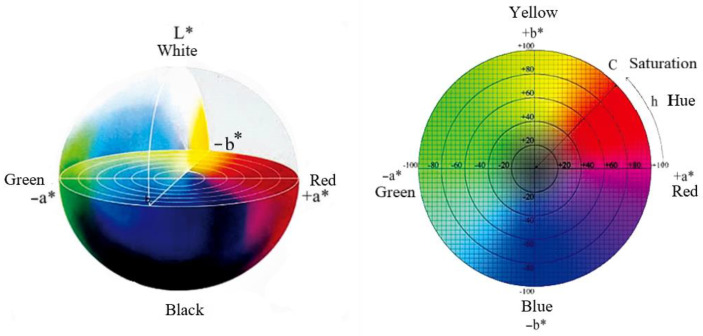
CIE Lab colour space.

**Figure 7 materials-15-07341-f007:**
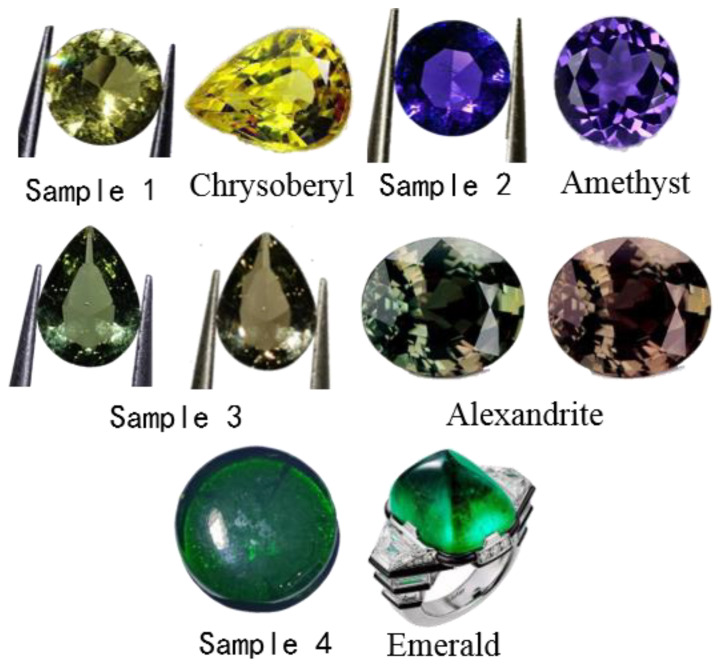
Comparison of imitation gemstone glass samples with natural gemstones.

**Figure 8 materials-15-07341-f008:**
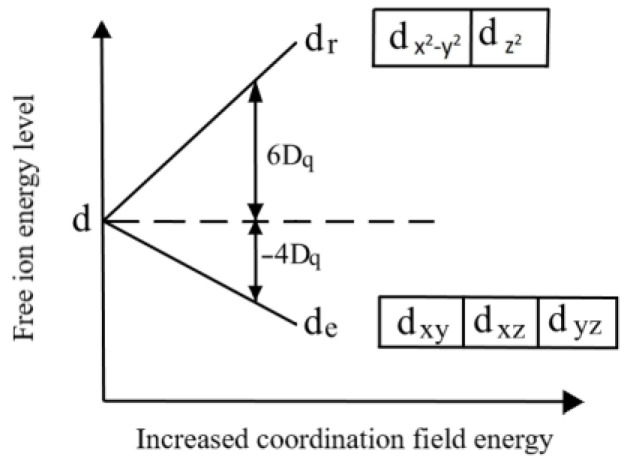
Splitting of 5 d orbital energies in the octahedral coordination field.

**Table 1 materials-15-07341-t001:** Ratio of raw materials to 4 types of imitation gemstone glass (Unit: mol%).

Glass Name	SiO_2_	K_2_CO_3_	NaNO_3_	Pr_6_O_11_	Nd_2_O_3_	Cr_2_O_3_	Na_2_B_4_O_7_·10H_2_O	Al_2_O_3_	ZnO
1 Chrysoberyl-imitation glass	56	11	25	0.35	0	0	1.30	4	1
2 Amethyst-imitation glass	56	11	25	0	1.90	0	1.30	4	1
3 Alexandrite-imitation glass	56	11	25	0.35	1.90	0	1.30	4	1
4 Emerald-imitation glass	56	11	25	0.35	1.90	0.50	1.30	4	1

**Table 2 materials-15-07341-t002:** Basic physical properties of imitation gemstone glass samples.

Sample	Colour		Refractive Index	Density ρ(g/cm^3^)	UV Fluorescence	
	D65 (6500 K)	A (2856 K)			LW 365 nm	SW 253.7 nm
1	Yellow		1.52	2.699	Weak (violet-red)	None
2	Blue-Purple		1.54	2.781	None	None
3	Yellow Green	Brownish red	1.56	2.875	Medium (violet-red)	None
4	Green		1.55	2.871	Medium (violet-red) (violet)Red	None

**Table 3 materials-15-07341-t003:** CIE chromaticity coordinates of samples.

Sample	Chromaticity Coordinates (6500 K)			Chromaticity Coordinates (2856 K)		
	R	G	B	R	G	B
1	39.8%	47.6%	12.6%	39.8%	47.6%	12.6%
2	29.8%	22.3%	47.9%	29.8%	22.3%	47.9%
3	37.3%	38.2%	24.5%	44.5%	33.6%	21.9%
4	25.7%	40.9%	33.4%	25.7%	40.9%	33.4%

**Table 4 materials-15-07341-t004:** Colour parameters of each sample in CIE Lab space.

Sample	Light Source	*L**	a*	b*	*C**	h_0_ (°)	Δ h_0_ (°)
1	D65	18.88	−3.54	7.62	8.40	114.89	17.35
	A	15.05	−1.23	9.28	9.36	97.54	
2	D65	23.32	2.03	−2.54	3.25	308.68	10.71
	A	14.32	2.27	−4.27	4.84	297.97	
3	D65	14.07	−7.20	16.18	17.71	113.99	45.42
	A	9.70	3.86	9.83	10.56	68.57	
4	D65	3.33	−0.29	−0.79	0.84	250.15	9.76
	A	3.85	0.73	−1.28	1.48	240.39	

## Data Availability

Not applicable.

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
