# Peer review of "Preparation and Basic Properties of Praseodymium-Neodymium-Chromium Containing Imitation Gemstone Glass"

_materials, 2022, doi:10.3390/ma15207341_

Round 1

Reviewer 1 Report

The manuscript presents the synthesis and characterization of praseodymium-chromium containing imitation gemstone glass. Although authors carried out several experimental works, the work has not shown any surprising observations or significant novelty. Therefore, I do believe the submitted work does not reach publication levels, and thus could not be recommended for publication in this journal.   

Author Response

Thank you so much for your review and comments. We are so glad to hear from you about the valuable suggestions and comments, which helped us improve our manuscript.

According to the comments we received, we have modified our manuscript carefully. All changes were highlighted in the text (in blue). Thanks again for your time and review. Look forward to hearing from you.

Reviewer 2 Report

This is well organized work with good motivation, the only minor in this work is that introduction section need to be flashed with some more references. then the work could be accpeted after minor revision 

Author Response

Dear Reviewer,

Thank you so much for your review and comments. We are so glad to hear from you about the valuable suggestions and comments, which helped us improve our manuscript.

According to the comments we received, we have modified our manuscript carefully. All changes were highlighted in the text (in blue). Thanks again for your time and review. Look forward to hearing from you.

Yours sincerely,

Siqi Zhang

Reviewer 3 Report

The manuscript requires major revision.

The title is incorrect because the glasses also contain neodymium.

Please rewrite Abstract. The details about the optical properties of samples 1 – 4 are meaningless because you do not explain the difference between the compositions of samples.

In the text of the manuscript, please place reference numbers into square brackets.

The list of references is insufficient. There are books devoted to glass coloration, see, i.e., W.A. Weyl, Coloured Glasses (Dawson's, London, 1959); T. Bates, Modern Aspects of the Vitreous State, Vol. 2 (Butterworths, London, 1962) ch. 5 p. 195. There are papers, on this matter, see, i.e., J.R. Johnson, Stained Glass and Imitation Gems, The Art Bulletin, Vol. 39, No. 3 (Sep., 1957), pp. 221-224, http://www.jstor.org/stable/3047715A.

There is no logic in introduction, it should be rewritten, and the aim of the present study should be clearly formulated.

Please rewrite Table 1. Please present the glass compositions either in weight or in mol%. Please check the sample 4 composition. It does not match your description.

Please rewrite the Preparation process. It should be the description of the procedure while you present the instruction.

Please explain how you prepared samples for recording of transmission spectra.

Please correct the figure 4 caption and discussion. You present not absorption but transmission spectra. Please assign absorption bands to certain electron transitions of rare-earth ions and Cr3+ ion.

Fig. 8 has no meaning and should be either excluded or substituted by the Tanabe-Sugano diagram for the Cr3+ ion in the octahedral ligand field.

Please prepare all references in the same style. Please delete the symbol [J], which you add to every reference.

Please correct typo in ref. [12].

In ref. [14], the author names are incorrect. Please correct the reference.

Author Response

(The authors gave the same response as above.)

Reviewer 4 Report

The authors are supposed to provide the transmission spectra of the pure (undoped) glass also for reference and calculate the CIE. 

It is suggested to present the graphs corresponding to the calculation of CIE  based on 6500 and 2856 K respectively. 

References are missing for different calculations/computational mechanisms in the manuscript. Those are to be updated carefully. 

The authors themselves noted that some references are not found means that the statements are baseless and through research has to be done. References are missing in large parts of discussion. 

Author Response

Dear Reviewer,

Thank you so much for your review and comments. We are so glad to hear from you about the valuable suggestions and comments, which helped us improve our manuscript again.

According to the comments we received, we have modified our manuscript carefully. All changes were highlighted in the text (in blue and red). Thanks again for your time and review. Look forward to hearing from you.

Yours sincerely,

Siqi Zhang, Keqing Li and Junyuan Pu

Corresponding Author: Siqi Zhang

Round 2

Reviewer 3 Report

Though the manuscript contains the results of interesting study, their presentation should be
sufficiently improved to warrant publication in Materials.

Author Response

(The authors gave the same response as above.)
